# Dysregulated *H19/Igf2* expression disrupts cardiac-placental axis during development of Silver-Russell syndrome-like mouse models

**Suhee Chang[1], Diana Fulmer[1,2], Stella K Hur[1], Joanne L Thorvaldsen[1], Li Li[1,2], Yemin Lan[1], Eric A Rhon-Calderon[1], Nicolae Adrian Leu[3], Xiaowen Chen[2], Jonathan A Epstein[1,2], Marisa S Bartolomei[1]\***

[1]Department of Cell and Developmental Biology, Epigenetics Institute, Perelman School of Medicine, University of Pennsylvania, Philadelphia, United States; [2]Penn Cardiovascular Institute, Perelman School of Medicine, University of Pennsylvania, Philadelphia, United States; [3]Department of Biomedical Sciences, School of Veterinary Medicine, Institute for Regenerative Medicine, University of Pennsylvania, Philadelphia, United States

**\*For correspondence:**
bartolom@pennmedicine.upenn.edu

**Competing interest:** The authors declare that no competing interests exist.

**Abstract** Dysregulation of the imprinted *H19/IGF2* locus can lead to Silver-Russell syndrome (SRS) in humans. However, the mechanism of how abnormal *H19/IGF2* expression contributes to various SRS phenotypes remains unclear, largely due to incomplete understanding of the developmental functions of these two genes. We previously generated a mouse model with humanized *H19/IGF2* imprinting control region (*hIC1*) on the paternal allele that exhibited *H19/Igf2* dysregulation together with SRS-like growth restriction and perinatal lethality. Here, we dissect the role of *H19* and *Igf2* in cardiac and placental development utilizing multiple mouse models with varying levels of *H19* and *Igf2*. We report severe cardiac defects such as ventricular septal defects and thinned myocardium, placental anomalies including thrombosis and vascular malformations, together with growth restriction in mouse embryos that correlated with the extent of *H19/Igf2* dysregulation. Transcriptomic analysis using cardiac endothelial cells of these mouse models shows that *H19/Igf2* dysregulation disrupts pathways related to extracellular matrix and proliferation of endothelial cells. Our work links the heart and placenta through regulation by *H19* and *Igf2*, demonstrating that accurate dosage of both *H19* and *Igf2* is critical for normal embryonic development, especially related to the cardiac-placental axis.

## Editor's evaluation

This important study took advantage of multiple mouse models with varying levels of H19 and Igf2 expression to dissect the role of H19 and Igf2 in cardiac and placental development. This work links the heart and placenta through regulation by H19 and Igf2, demonstrating that an accurate dosage of both H19 and Igf2 is critical for normal embryonic development, especially related to the cardiac-placental axis. The topic is of significance, and the data are of high quality and convincing.

## Introduction

Genomic imprinting is a mammalian-specific phenomenon where a small number of genes are expressed in an allele-specific manner. Functionally, imprinted genes have central roles in development

Figure 1. *H19/Igf2* cluster and mouse models utilized in this study. (**A**) A schematic representation of the wild-type *H19/Igf2* cluster in mouse. The maternally expressed *H19* and the paternally expressed *Igf2* genes are shown in red and blue, respectively. Black lollipops on the paternal allele represent DNA methylation. The maternal imprinting control region (ICR) is bound to CTCF proteins (pink hexagons) at CTCF-binding sites, forming an insulator that blocks the maternal *Igf2* promoter from the shared enhancers (green circles). These enhancers interact with the *H19* promoter on the maternal allele and *Igf2* promoter on the paternal allele. (**B**) Schematic of the mouse endogenous *H19/Igf2* ICR, *H19^{hIC1}* (shortened as hIC1; **Hur et al., 2016**), *H19^{Δ2.8kb-H19}* (shortened as Δ H19), and *H19^{Δ3.8kb-5'H19}* (shortened as Δ 3.8; **Thorvaldsen et al., 2002**; **Thorvaldsen et al., 2006**) alleles. Restriction site locations (kb) are relative to the *H19* transcription start site. Gray lines on the ICR represent conserved CTCF-binding sequences.

and growth in both humans and mice (**Barlow and Bartolomei, 2014**). Additionally, proper gene dosage of most imprinted genes is essential for normal development. Human chromosome 11 and the orthologous region on mouse chromosome 7 harbor two jointly controlled growth regulators with opposing functions; *H19* long non-coding RNA (lncRNA) and Insulin-like Growth Factor 2 (*IGF2/Igf2*). These two imprinted genes share an imprinting control region (ICR), a *cis*-regulatory element located between two genes, which is essential for their allele-specific expression, as well as tissue-specific enhancers located downstream of *H19* (**Chang and Bartolomei, 2020**). The *H19/IGF2* ICR, which is designated as IC1 in humans, binds CTCF on the maternal allele, forming an insulator and enabling *H19* exclusive access to the shared enhancers (**Figure 1A**). On the paternal allele, the *H19/IGF2* ICR is methylated, which prevents CTCF from binding and an insulator from forming. Consequently, *IGF2* usurps the shared enhancers, and *H19* is repressed on the paternal allele. Ultimately, allele-specific ICR methylation facilitates monoallelic expression of *H19* and *IGF2* with *H19* expressed from the maternal allele and *IGF2* expressed from the paternal allele.

Dysregulation of the *H19/IGF2* cluster is associated with two growth disorders, Beckwith-Wiedemann syndrome (BWS) and Silver-Russell syndrome (SRS). In contrast to overgrowth observed for BWS, SRS is characterized by intrauterine growth restriction resulting in small for gestational age births. Other symptoms of SRS vary widely among patients and include hemihypotrophy, cognitive impairment, relative macrocephaly, and fifth-finger clinodactyly (**Wakeling et al., 2017**). Approximately, 50% of patients with SRS exhibit IC1 hypomethylation (**Eggermann et al., 2011**). Lack of methylation may allow the formation of an ectopic insulator on paternal IC1, which likely explains why this class of SRS individuals has biallelic *H19* expression and greatly diminished *IGF2* expression (**Abi**

*Habib et al., 2017*; *Gicquel et al., 2005*). Importantly, ICR mutations in the mouse that were generated to study imprinted gene regulation provided critical information, suggesting a role for *H19* and *IGF2* in SRS. For example, mutating CpGs at CTCF sites on the paternal *H19/Igf2* ICR resulted in the loss of ICR methylation, activation of paternal *H19*, and reduced *Igf2* expression (*Engel et al., 2004*). This mutation led to restricted embryonic growth, which phenocopies SRS. Nevertheless, although we and others successfully modeled a subset of SRS mutations in the mouse, not all mutations were translatable because the mouse *H19/Igf2* ICR lacks extensive sequence conservation with human IC1. Thus, a mouse model with human IC1 sequence substituted for the endogenous mouse *H19/Igf2* ICR was generated (*H19^hIC1^*, shortened as *hIC1*; *Hur et al., 2016*) to model human mutations more precisely (*Freschi et al., 2018*; *Freschi et al., 2021*). Consistent with expectations, maternally transmitted *hIC1* successfully maintained insulator function, suggesting the possibility to model human IC1 mutations endogenously in mice upon maternal transmission. In contrast, paternally transmitted *hIC1* showed loss of methylation and formation of an ectopic insulator. Consequently, paternal *hIC1* transmission caused elevated *H19* expression and *Igf2* depletion together with growth restriction and embryonic lethality (*Hur et al., 2016*). Although the epigenetic defects and growth restriction of these mice nicely model SRS symptoms, many SRS individuals are viable. A potential explanation for such discrepancy between human and mouse could reflect the mosaic nature of epigenetic defects in human (*Soellner et al., 2019*), with a subset of cells showing normal methylation patterns.

The mechanism by which *H19/IGF2* expression dysregulation causes SRS phenotypes is unknown, largely because the function of these two genes during development is incompletely understood. *IGF2* is a well-described growth factor promoting fetoplacental growth, which functions in an endocrine/paracrine manner through binding to IGF/insulin receptors (*Harris and Westwood, 2012*). Decreased *IGF2* levels in patients with SRS suggest that IGF2 contributes to restricted growth in these patients (*Abi Habib et al., 2017*; *Begemann et al., 2015*; *Gicquel et al., 2005*). Consistently, in mice, paternal-specific deletion of *Igf2* resulted in loss of *Igf2* expression and pre- and postnatal growth restriction (*DeChiara et al., 1991*; *Haley et al., 2012*). In contrast, the exact role of *H19* remains unclear. Previous studies in mouse suggested that *H19* lncRNA is a precursor for *microRNA 675* (*Mir675*), which regulates *Igf1r* expression (*Keniry et al., 2012*) and that *H19* represses *Igf2* expression in trans (*Gabory et al., 2009*). As a result, *H19* has been largely overlooked and suggested to be an occasional regulator of *Igf2* expression. Nevertheless, a previously described mouse model with *H19* overexpression without changes in *Igf2* expression showed embryonic growth restriction (*Drewell et al., 2000*), suggesting that *H19* mediates growth suppression independently from *Igf2*. Here, we report severe developmental defects of the heart and placenta in mouse models with dysregulated *H19/Igf2* expression. Embryos with the paternal *hIC1* showed atrioventricular (AV) cushion defects in the heart coupled with ventricular septal defects (VSDs) and extremely thinned myocardial walls. Combined with placental anomalies, the cardiac defects most likely contribute to the lethality of these mice (*Kochilas et al., 1999*; *Snider and Conway, 2011*). Deletion of *H19* from the maternal allele, thereby reducing *H19* levels, failed to rescue completely the lethality and growth restriction associated with the paternal inheritance of *hIC1*. A minimal rescue of the earlier growth and resorption frequency was, however, observed with normalized *H19* expression. Ultimately, modifying both *H19* and *Igf2* expression was necessary to rescue most phenotypes. Transcriptomic analysis of embryonic cardiac endothelial cells identified several key signaling pathways that are affected by the dysregulated *H19* and *Igf2* and are potentially responsible for the paternal *hIC1*-related cardiac defects. This work emphasizes the importance of accurate dosage of *H19* and *Igf2* expression in normal cardiac and placental development, disruption of which can lead to SRS-like pathologies.

## Results

Mouse models with genetic modifications that perturb *H19* and *Igf2* to different extents were used to address the phenotypic consequences of abnormal *H19* and *Igf2* levels (*Figure 1B*). *hIC1* refers to the humanized *H19^hIC1^* allele that substitutes the endogenous mouse *H19/Igf2* ICR with the corresponding human IC1 sequence, which was initially generated to study human BWS and SRS mutations in the mouse (*Freschi et al., 2018*; *Hur et al., 2016*). Paternal transmission of *hIC1* (+/*hIC1*) was previously reported to increase *H19* and greatly diminish *Igf2* expression and resulted in embryonic lethality.

## Cardiac and placental defects in *+/hIC1* embryos

Our initial experiments examined the developmental phenotype of *+/hIC1* embryos to determine how abnormal *H19/Igf2* expression resulted in dramatic growth defects and lethality. As previously reported (*Hur et al., 2016*), *+/hIC1* embryos showed severe growth restriction (note that for heterozygous embryos, maternal allele is written first, *Figure 2A*). The growth restriction appeared as early as E11.5 and was greatly exaggerated by the end of gestation. Although E18.5 *+/hIC1* embryos were observed alive and weighed approximately 40% of their wild-type littermates, *+/hIC1* neonates were perinatally lethal, with no live pups found on the day of birth. To ascertain the source of lethality, we first examined lungs from dead *+/hIC1* neonates. The lungs floated in water, demonstrating that *+/hIC1* pups respired after birth (*Borensztein et al., 2012*). Additionally, *+/hIC1* neonates did not have a cleft palate, but no milk spots were found in their abdomen, indicating a lack of feeding.

Histological analyses were performed throughout development on major organs where *H19* and *Igf2* are highly expressed. Severe developmental defects were found in the *+/hIC1* heart and placenta. Cardiac defects in *+/hIC1* embryos were observed as early as E12.5, where the superior and inferior endocardial cushions failed to fuse into a common AV cushion (*Figure 2C*). The cushion defect preceded incomplete interventricular septum (IVS) formation. At E15.5, a severe perimembranous VSD was observed in all five *+/hIC1* hearts that were evaluated (*Figure 2C*). Importantly, this congenital heart defect resembles malformations reported in several SRS patients with IC1 hypomethylation (*Ghanim et al., 2013*), although the prevalence is unclear and may reflect the degree of mosaicism in SRS. Additionally, E15.5 hearts showed an extremely thin myocardium (*Figure 2C* and *Figure 2— figure supplement 1A*). Both VSD and thinned ventricular walls persisted in E17.5 *+/hIC1* hearts (*Figure 2B and C*). Additionally, 6 out of 16 *+/hIC1* hearts in the late gestation group (E15.5 to P0) had bicuspid pulmonary valve, a rare cardiac defect in which the pulmonary valve only develops two cusps as opposed to the normal tricuspid structure (*Figure 2D* and *Figure 2—figure supplement 1B*). These results demonstrate that paternal *hIC1* transmission results in variably penetrant cardiac phenotypes. Notably, AV valvuloseptal morphogenesis, the fusion of the AV cushion and nascent septa during cardiogenesis, is required for proper cardiac septation (*Eisenberg and Markwald, 1995*). Segmentation of the heart into four separate chambers is required to establish distinct pulmonary and systemic blood flow during heart development and is required to prevent the mixing of oxygenated and deoxygenated blood. We speculate that the failure of *hIC1* mutants to establish complete ventricular septation could have led to a reduced ability to provide oxygen and nutrient rich blood to the rest of the developing body (*Savolainen et al., 2009*; *Spicer et al., 2014*).

Another major organ with high *H19/Igf2* expression, which forms early in development, is the placenta. Multiple developmental defects were observed in *+/hIC1* placentas. As previously described (*Hur et al., 2016*), *+/hIC1* placentas were growth restricted throughout development (*Figure 3A*), although the fetal to placental weight ratio was not affected through E15.5 (*Figure 3—figure supplement 1A*). However, at E17.5, the fetal to placental weight ratio was lower in *+/hIC1* conceptuses, indicating that the fetal growth restriction was more severe than the placental growth restriction as the concepti neared term (*Figure 3B*). In addition to placental undergrowth, the junctional to labyrinth zone ratio was increased in *+/hIC1* placentas (*Figure 3C* and *Figure 3—figure supplement 1B*), suggesting that the growth of the labyrinth layer, where the maternal-fetal exchange occurs, was more affected. Moreover, *H19* overexpression was exaggerated in the labyrinth in E17.5 *+/hIC1* placentas, while the *Igf2* depletion was consistent throughout the whole placenta (*Figure 3D*), possibly indicating that *H19* overexpression contributed disproportionately to the phenotype of growth restriction in the labyrinth. Large thrombi were observed in the labyrinth zone of these *+/hIC1* placentas (*Figure 3F*), in a male-skewed manner (*Figure 3E*). As the thrombi could be formed due to defective vasculature structures, wild-type and *+/hIC1* placentas were stained for CD34, a marker for the fetoplacental endothelial cells that line the microvessels in the labyrinth layer. Vessels in the *+/hIC1* labyrinth were highly dilated (*Figure 3G*), and quantification of the stained areas showed that the microvascular density was significantly decreased in *+/hIC1* placentas among males (*Figure 3H* and *Figure 3— figure supplement 1C*). Although previous studies reported that *Igf2*-null mice had lower placental glycogen concentration (*Lopez et al., 1996*) and *H19* deletion led to increased placental glycogen storage (*Esquiliano et al., 2009*), Periodic acid-Schiff (PAS) staining on *+/hIC1* placentas showed that the glycogen content is not significantly different between wild-type and *+/hIC1* placentas (*Figure 3— figure supplement 1D*). The reduced labyrinth layer and defective microvascular expansion likely

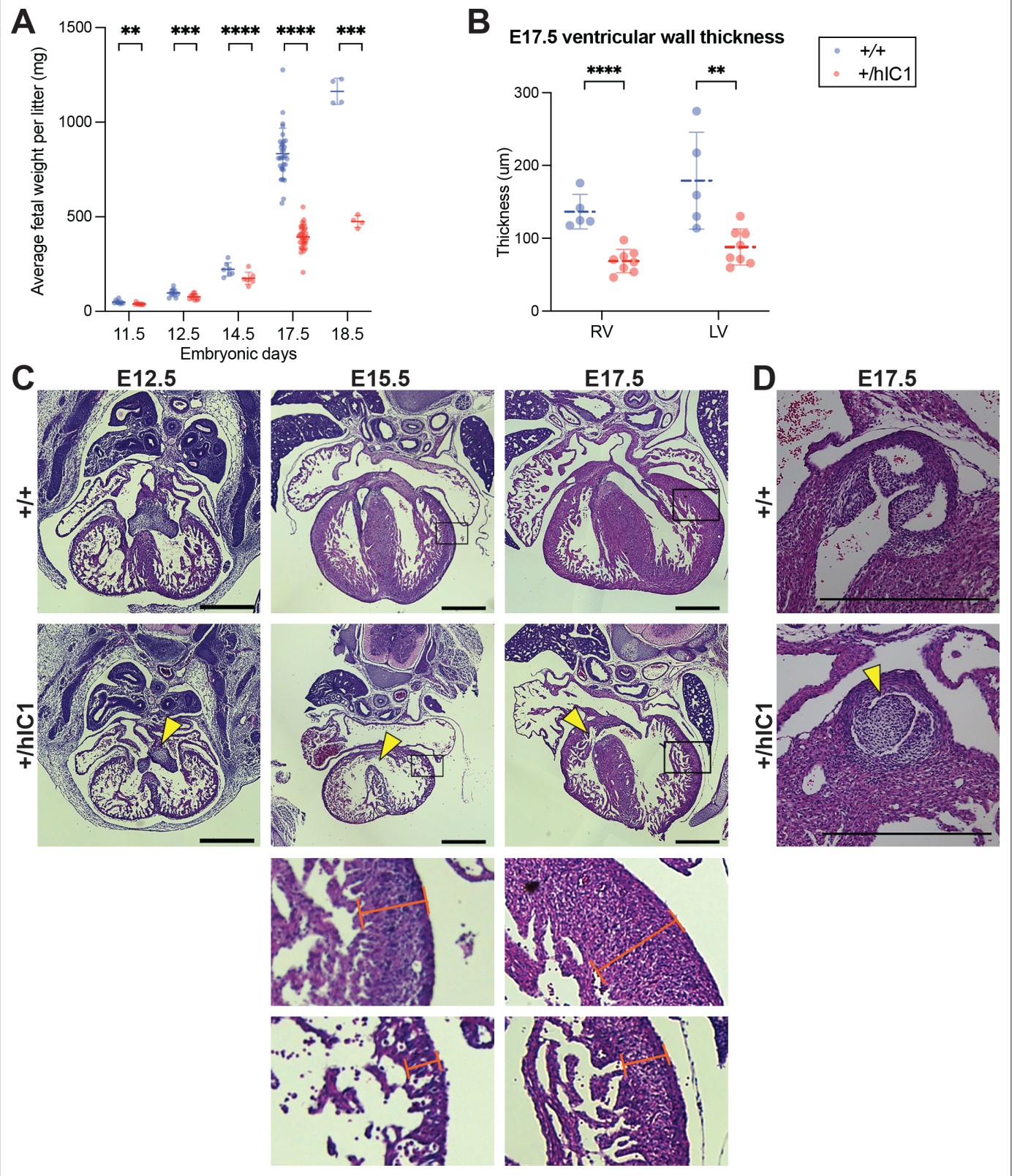

**Figure 2.** Growth anomalies and cardiac defects of *+/hIC1* embryos. (**A**) Fetal weight of the wild-type (blue) and *+/hIC1* (red) embryos at E11.5, E12.5, E14.5, E17.5, and E18.5 (mean ± SD). Each data point represents an average weight of each genotype from one litter. 8 litters for E11.5, 10 litters for E12.5, 7 litters for E14.5, 31 litters for E17.5, and 4 litters for E18.5 are presented. (**B**) Quantification of ventricular wall thickness (μm), measured from E17.5 wild-type and *+/hIC1* hearts (mean ± SD). Each data point represents an individual conceptus. Five wild-type and eight *+/hIC1* embryos from

*Figure 2 continued on next page*

Figure 2 continued

four different litters were examined. (**C**) Representative cross-sections of wild-type and *+/hIC1* embryonic hearts at E12.5, E15.5, and E17.5, stained with hematoxylin and eosin. All the hearts represented here are from female fetuses. Note the lack of fusion between atrioventricular (AV) cushions at E12.5, the ventricular septal defect (VSD) at E15.5, and E17.5 in *+/hIC1* hearts (yellow arrowheads). The boxed regions of E15.5 and E17.5 images are enlarged at the bottom of the figure, to show where the ventricular wall thickness is measured. Scale bars = 500 μm. (**D**) A representative image of pulmonary valves in E17.5 wild-type and *+/hIC1* hearts. The *+/hIC1* right pulmonary cusp is enlarged (yellow arrowhead), and there is no cusp in the anterior position, in contrast to the tricuspid structure in the wild-type heart. Scale bars = 500 μm. Statistics used are (**A**) multiple paired t-test and (**B**) multiple unpaired t-test with **p<0.01, ***p<0.001, ****p<0.0001, and ns = not significant.

The online version of this article includes the following figure supplement(s) for figure 2:

**Figure supplement 1.** Supplementary data for anomalies observed in *+/hIC1* hearts.

compromised the ability of *+/hIC1* placentas to supply nutrients and oxygen to the fetus. These results support the hypothesis that the abnormal growth of *+/hIC1* embryos may be explained by failure in multiple organs, especially the heart and placenta, which are developmentally linked (***Barak et al., 2019***).

Additionally, we examined expression of the *H19*-derived *Mir675* in *+/hIC1* placentas to determine if the level of *Mir675* is correlated with changes in *H19* and *Igf2* expression. This analysis was conducted at E15.5 when cardiac and placental defects were already observed in *+/hIC1* embryos (***Figure 2—figure supplement 1A***, B and ***Figure 3—figure supplement 1B***). Despite increased *H19* expression and *Igf2* depletion (***Hur et al., 2016***), *Mir675* was not significantly increased in *+/hIC1* placentas compared to the wild-type (***Figure 3—figure supplement 1E***). Thus, we conclude that the placental phenotypes observed in *+/hIC1* mice are solely attributable to the increased *H19* lncRNA, irrespective of *Mir675.* Another possibility is that the disproportionate *H19* overexpression in the labyrinth layer at E17.5 (***Figure 3D***) was also present at E15.5 because growth suppression was more severe in labyrinth than in junctional zone in E15.5 (***Figure 3—figure supplement 1B***) and in E17.5 placenta (***Figure 3C***). This would have made it difficult to detect a substantial difference in *Mir675* expression in the whole placenta.

## Normalizing *H19* expression partially rescues paternal *hIC1* defects

It has been previously reported that *Igf2* null mice are viable (***DeChiara et al., 1991***). Thus, we hypothesize that *H19* overexpression combined with a loss of *Igf2* expression is the molecular contributor to the lethality of paternal *hIC1* transmission. To examine if reduced *H19* expression would rescue the paternal *hIC1* transmission phenotypes, we generated a mouse model with deletion of the *H19* transcription unit (*H19^{Δ2.8kb-H19}*; shortened as ΔH19; ***Figure 1B***, ***Figure 4—figure supplement 1A and B***). Consistent with previous reports, maternal deletion of *H19* (ΔH19/+) led to an absence of *H19* expression and tissue-specific minimal activation of maternal *Igf2* (***Figure 4—figure supplement 1C***). These mice are viable and fertile, regardless of whether the deletion is maternally or paternally transmitted, although maternal transmission is associated with increased fetal weight from E14.5 and onward (***Figure 4—figure supplement 1D***).

Heterozygous ΔH19 females were mated with heterozygous *hIC1* males to generate ΔH19/hIC1 embryos. These embryos were expected to have lower *H19* expression compared to *+/hIC1* embryos, as the maternal *H19* expression was silenced (***Figure 4A***). Among four possible genotypes from this breeding, *+/hIC1* embryos constituted approximately 15% per litter at E17.5, as opposed to the expected Mendelian ratio of 25%. In contrast, ΔH19/hIC1 embryos comprised approximately 30% per litter at E17.5, indicating partial rescue of the resorption frequency by maternal *H19* deletion (***Figure 4B*** and ***Figure 4—figure supplement 2A***). However, ΔH19/hIC1 embryos still exhibited perinatal lethality, with no live pups observed on the day of birth. With respect to growth restriction, the maternal ΔH19 allele partially rescued the phenotype. At E11.5, ΔH19/hIC1 fetal weight was not significantly different from wild-type littermates (***Figure 4C***). However, at late gestation (E17.5), although ΔH19/hIC1 fetuses had a significant increase in fetal weight compared to the *+/hIC1* fetuses, ΔH19/hIC1 fetuses remained significantly smaller compared to the wild-type. As perinatal lethality was still observed, conceptuses were analyzed histologically to characterize their developmental defects. The AV cushion defect persisted in all examined E13.5 ΔH19/hIC1 embryos, and 50% of the examined E17.5 ΔH19/hIC1 hearts showed either perimembranous or muscular VSDs (***Figure 4D***). Thrombi were still present in approximately 50% of ΔH19/hIC1 placentas (***Figure 4E*** and ***Figure 4—figure***

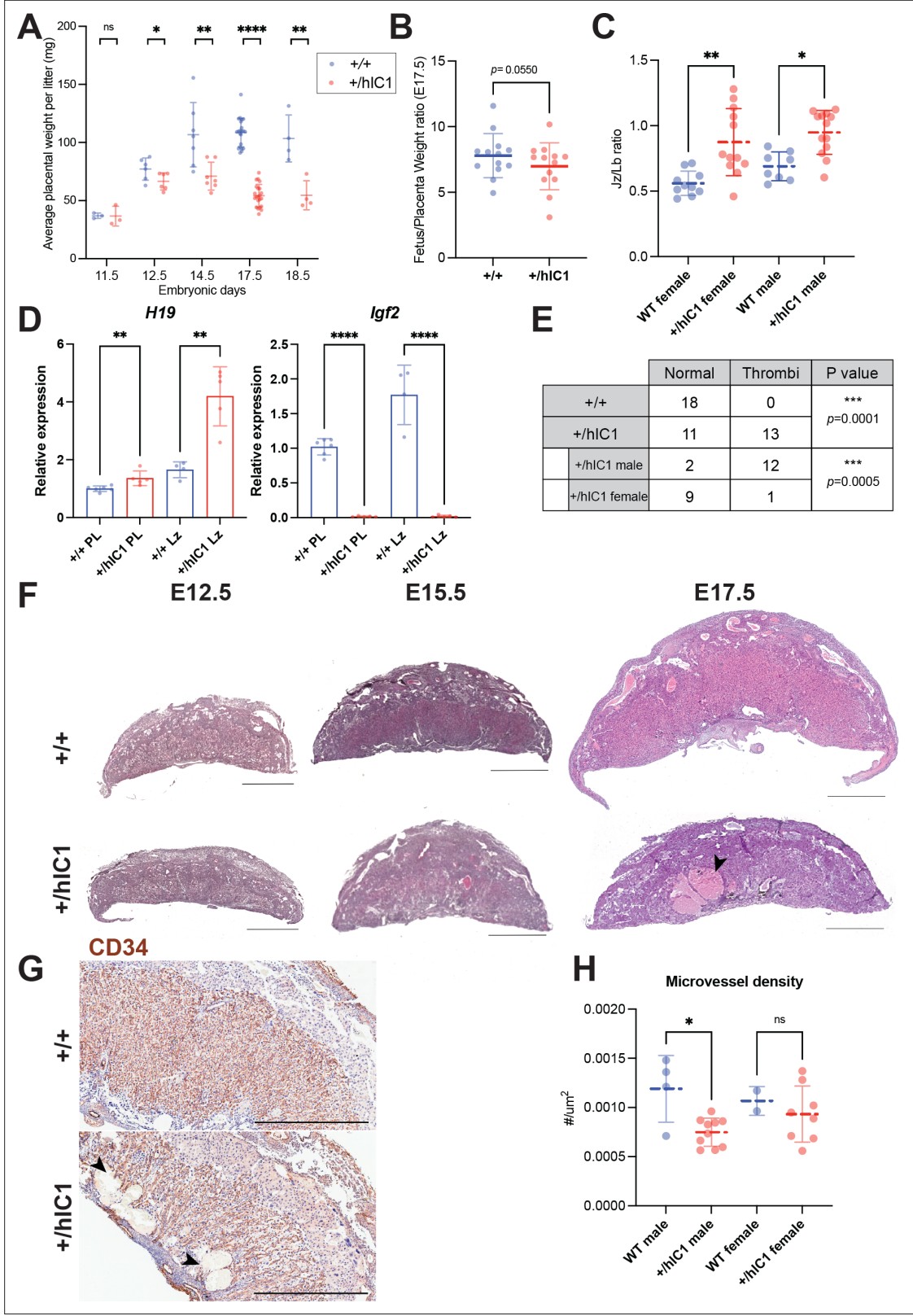

**Figure 3.** Placental defects of *+/hIC1* embryos. (**A**) Placental weight of the wild-type (blue) and *+/hIC1* (red) samples at E11.5, E12.5, E14.5, E17.5, and E18.5 (mean ± SD). Each data point represents an average weight of each genotype from one litter. 3 litters for E11.5, 6 litters for E12.5, 7 litters for E14.5, 22 litters for E17.5, and 4 litters for E18.5 are presented. (**B**) Fetal to placental weight ratios in E17.5 wild-type and *+/hIC1* samples (mean ± SD). Each data point represents the average F/P ratio of each genotype from one litter. 13 litters are presented. (**C**) Junctional zone (Jz) to labyrinth

*Figure 3 continued on next page*

**Figure 3 continued**

(Lb) ratio in E17.5 wild-type and *+/hIC1* placentas (mean ± SD). (**D**) Relative total expression of *H19* and *Igf2* in E17.5 wild-type and *+/hIC1* placentas and Lb samples (mean ± SD). (**E**) Number of wild-type, male, and female *+/hIC1* placentas with thrombi observed. (**F**) Representative cross-sections of E12.5, E15.5, and E17.5 wild-type and *+/hIC1* placentas stained with hematoxylin and eosin. All depicted E12.5 and E15.5 placentas are female. The E17.5 wild-type placenta is female, and *+/hIC1* placenta is male. Black arrowhead indicates a large thrombus formed in the *+/hIC1* Lb. Scale bars = 1 mm. (**G**) Representative images of CD34 immunostaining counterstained with hematoxylin on E17.5 wild-type female and *+/hIC1* male placental sections. Black arrowheads indicate thrombi in the *+/hIC1* Lb. Scale bars = 1 mm. (**H**) Quantification of the microvessel density in E17.5 wild-type and *+/hIC1* placentas. 4 wild-type male, 10 *+/hIC1* male, 2 wild-type female, 8 *+/hIC1* female placentas from six litters were quantified. (**C, D, and H**) Each data point represents an individual conceptus from different litters. Statistics used are (**A and B**) multiple paired t-test, (**C**) one-way ANOVA with Tukey's multiple comparisons test, (**D and H**) multiple unpaired t-test, (**E**) Fisher's exact test. *p<0.05, **p<0.01, ***p<0.001, ****p<0.0001, and ns = not significant.

The online version of this article includes the following figure supplement(s) for figure 3:

**Figure supplement 1.** Supplementary data for anomalies observed in *+/hIC1* placentas.

supplement 2D), and placental weight remained significantly lower compared to wild-type littermates (**Figure 4—figure supplement 2B**). Of note, none of these histological defects were observed in *ΔH19/+* embryonic hearts and placentas (see **Figure 4D** for example; three *ΔH19/+* hearts each from E13.5, E15.5, and E17.5 concepti, and 20 E17.5 *ΔH19/+* placentas were examined). E17.5 *ΔH19/hIC1* tissues demonstrated wild-type levels of *H19* expression, while the *Igf2* expression remained markedly lower than wild-type (**Figure 4F** and **Figure 4—figure supplement 2C**). From these results, we conclude that restoring *H19* expression is not sufficient to rescue completely the lethality and developmental defects upon paternal transmission of *hIC1*. Thus, phenotypes are likely caused by abnormal expression of both *H19* and *Igf2*.

## Paternal *hIC1* defects are rescued by deletion of the maternal *H19/Igf2* ICR

Finally, we utilized a previously published mouse model with a 3.8 kb deletion spanning the *H19/Igf2* ICR (*H19^{Δ3.8kb-5'H19}*; shortened as *Δ3.8*) to modulate both *H19* and *Igf2* expression (**Thorvaldsen et al., 2002**; **Thorvaldsen et al., 2006**). Absence of the maternal *H19/Igf2* ICR activates the maternal *Igf2* allele and reduces *H19* expression. Thus, *Igf2* expression from the maternal allele in *Δ3.8/hIC1* embryos was expected to restore *Igf2* levels (**Figure 5A**).

Crosses between heterozygous *Δ3.8* females and heterozygous *hIC1* males produced the expected Mendelian ratio of offspring with *Δ3.8/hIC1* mice appearing fully viable. Both fetal and placental weights were not significantly different between *Δ3.8/hIC1* and wild-type concepti at E17.5 (**Figure 5B** and **Figure 5—figure supplement 1A**), demonstrating full rescue of the lethality and growth restriction. However, a subset of *Δ3.8/+* and *Δ3.8/hIC1* embryonic hearts (three out of six *Δ3.8/+* hearts and two out of five *Δ3.8/hIC1* hearts) had VSDs, although the lesions in the IVS were smaller than those found in *+/hIC1* hearts (**Figure 5C**). While *Igf2* expression in the E17.5 *Δ3.8/hIC1* hearts was restored to wild-type levels, normalization of *H19* expression varied among embryos. Although not statistically significant in heart, *Δ3.8/hIC1* embryos tended to have higher *H19* expression compared to the wild-type littermates (**Figure 5D** and **Figure 5—figure supplement 1B**). These results suggest that the physiological levels of *H19* and *Igf2* expression are critical for normal cardiac development, and the variability in *H19* rescue could help to explain the varying penetrance of the cardiac phenotype. No thrombi were detected in the *Δ3.8/hIC1* placentas, and the placental morphology was normal with the junctional to labyrinth zone ratio not significantly different compared to wild-type (**Figure 5—figure supplement 1C**). In sum, restoring both *H19* and *Igf2* to near wild-type levels was necessary for the full rescue of the most severe pathologies of paternal *hIC1* transmission.

## Transcriptomic analysis of cardiac endothelial cells with various *H19/Igf2* expression

Severe cardiac phenotypes associated with paternal *hIC1* transmission prompted us to question the mechanism of how dysregulated *H19/Igf2* causes such developmental defects. The paternal *hIC1*-associated cardiac phenotypes were observed as early as E12.5 when AV cushion fusion is delayed in developing hearts (**Figure 2B**), making E12.5 an optimal time point to identify the key pathways of valve development and cardiac septation that are disrupted by *H19/Igf2* dysregulation. Endothelial and endothelial-derived cells comprise the majority population in the AV cushion and majority

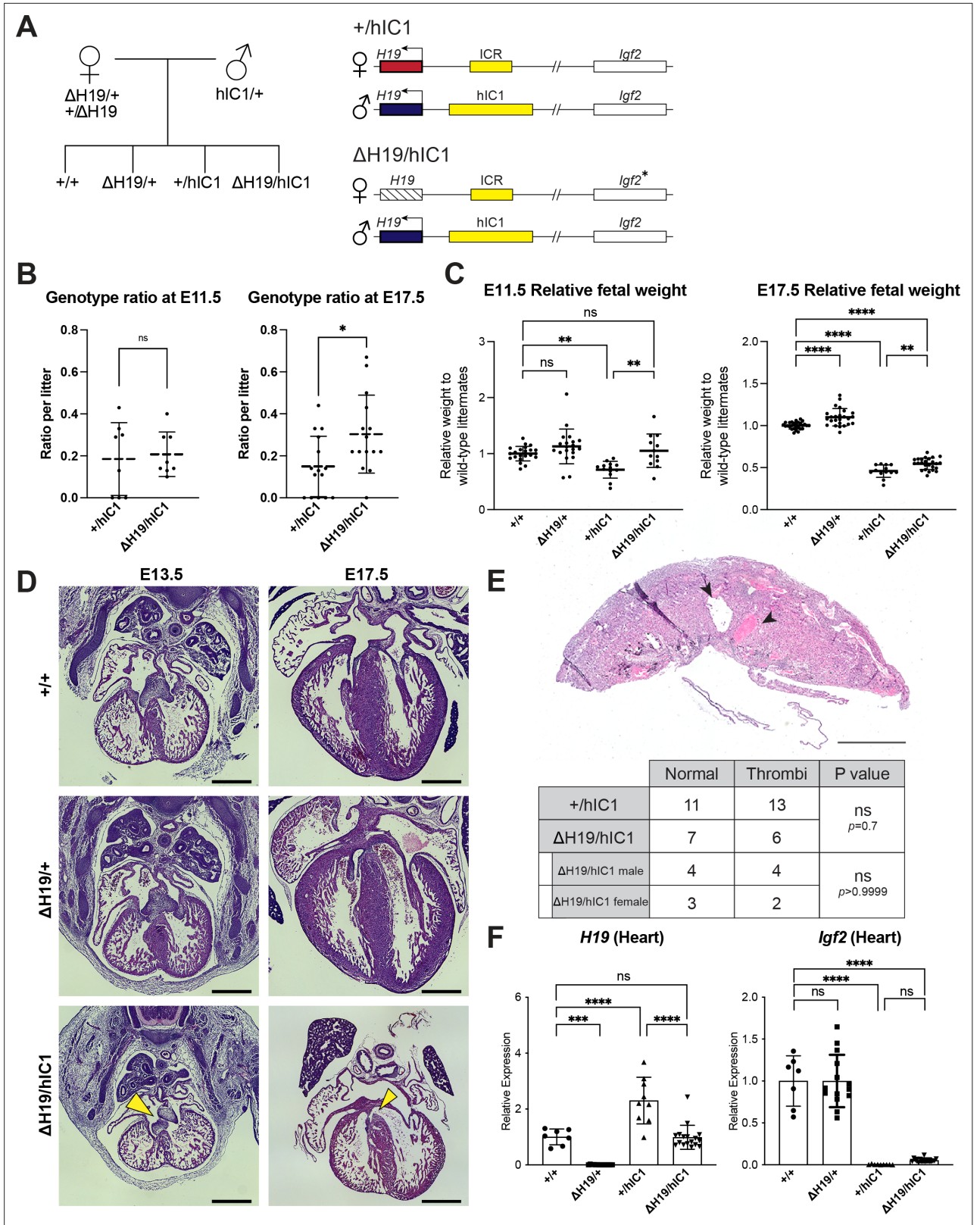

**Figure 4.** Normalizing *H19* expression through the maternal deletion of *H19*. (**A**) A schematic representation of the rescue breeding between ᐃ*H19* heterozygous female and *hIC1/+* male mice. ᐃ*H19/hIC1* embryos are expected to express *H19* only from the paternal allele and maternally express *Igf2* in a tissue-specific manner. (**B**) Ratio of *+/hIC1* and ᐃ*H19/hIC1* embryos observed in E11.5 and E17.5 litters (mean ± SD). 8 E11.5 litters and 15 E17.5 litters with litter size larger than five pups were examined. Each data point represents one litter. (**C**) Relative fetal weights of wild-type, ᐃ*H19/+*, *+/*

*Figure 4 continued on next page*

*Figure 4 continued*

*hIC1*, and *△H19/hIC1* embryos at E11.5 and E17.5, normalized to the average body weight of the wild-type littermates (mean ± SD). (**D**) Representative cross-sections of wild-type, *△H19/+* and *△H19/hIC1* embryonic hearts at E13.5 and E17.5, stained with hematoxylin and eosin. Note the cushion defect at E13.5 and the ventricular septal defect (VSD) at E17.5 in *△H19/hIC1* hearts (yellow arrows). All E13.5 samples and E17.5 wild-type sample are male, and E17.5 *△H19/+* and *△H19/hIC1* samples are female. Scale bars = 500 μm. (**E**) Top: Representative cross-section of E17.5 *△H19/hIC1* male placenta stained with hematoxylin and eosin. Black arrowheads indicate thrombi. Scale bar = 1 mm. Bottom: Number of the wild-type, male, and female *+/hIC1* placentas with thrombi observed. (**F**) Relative total expression of *H19* and *Igf2* in E17.5 wild-type, *△H19/+*, *+/hIC1*, and *△H19/hIC1* hearts (mean ± SD). (**C and F**) Each data point represents an individual conceptus from different litters. Statistics used are (**B, C, and F**) one-way ANOVA with Tukey's multiple comparisons test and (**E**) Fisher's exact test. *$p<0.05$, **$p<0.01$, ***$p<0.001$, ****$p<0.0001$, and ns = not significant.

The online version of this article includes the following source data and figure supplement(s) for figure 4:

**Figure supplement 1.** Characterization of *ΔH19* allele.

**Figure supplement 1—source data 1.** Raw gel images for *Figure 4—figure supplement 1B*.

**Figure supplement 1—source data 2.** Raw gel images for *Figure 4—figure supplement 1C*.

**Figure supplement 1—source data 3.** Raw gel images for *Figure 4—figure supplement 1C*.

**Figure supplement 2.** Supplementary data for rescue upon maternal *ΔH19* transmission.

non-myocyte population of the ventricular septum at E12.5 (*von Gise and Pu, 2012*). Moreover, both *H19* and *Igf2* are strongly expressed in the endocardial layer of developing heart (*García-Padilla et al., 2019*). Therefore, transcriptomic analysis was performed on cardiac endothelial cells of each mutant.

CD31+ cardiac endothelial cells from E12.5 wild-type, *+/hIC1*, *ΔH19/+*, *ΔH19/hIC1*, *Δ3.8/+*, and *Δ3.8/hIC1* embryos were collected for RNA sequencing. First, we confirmed that these six groups show gradual alteration of *H19/Igf2* expression (*Figure 6A*), which enabled us to generate multiple comparisons relative to *H19* and *Igf2* levels and potentially enabling attribution of phenotypes to *H19* or *Igf2*. Notably, *H19/Igf2* expression was indistinguishable in the wild-type and *Δ3.8/hIC1* samples. Additionally, there were no sex-specific differences in *H19/Igf2* expression across all the groups.

To elucidate candidate genes and cellular processes in conferring the paternal *hIC1*-specific cardiac phenotypes, we compared *+/hIC1* and wild-type samples. This comparison resulted in 224 significant differentially expressed genes (DEGs; *Figure 6B*, left). Gene ontology (GO) analysis revealed that pathways related to proliferation and remodeling of endothelial and epithelial cells are enriched in these DEGs (*Figure 6B*, right; *Mi et al., 2019*). Notably, genes involved in extracellular matrix (ECM) and specifically, collagen matrix production, were differentially enriched in E12.5 *+/hIC1* samples; consistent with increased collagen in the E17.5 *+/hIC1* hearts, as confirmed through immunofluorescence staining (*Figure 6C*, *Figure 6—figure supplement 1A and B*). Thus, differential gene expression in the presence of paternal *hIC1* was evidenced by E12.5 and correlated with histological consequences persisting through gestation.

Because cardiac phenotypes were indistinguishable between males and females, we next used same-sex comparisons between *+/hIC1* and wild-type to identify the *hIC1*-specific DEGs that are significant in both males and females (*Figure 6D*, left and *Figure 6—figure supplement 1C*). The 25 genes that overlapped between male and female *+/hIC1*-specific DEGs included *Col14a1*, which was upregulated in *+/hIC1*, emphasizing the importance of collagen-related ECM in *hIC1*-associated cardiac defects. The expression pattern of these 25 genes across all samples clustered *+/hIC1* and *ΔH19/hIC1*, the two groups with the most severe cardiac defects and perinatal lethality (*Figure 6D*, right). Although the number of *+/hIC1*-specific DEGs largely differed between males and females, there was no sex-specific bias on the X chromosome (*Figure 6—figure supplement 1D*).

To separate the effect of *Igf2* depletion from that of *H19* overexpression, we utilized *ΔH19/hIC1* samples. Consistent with previous observations from E17.5 hearts (*Figure 4F*), *ΔH19/hIC1* endothelial cells exhibited low *Igf2*, but *H19* expression was not significantly different from wild-type (*Figure 6—figure supplement 2A*). Thus, *ΔH19/hIC1* samples were compared to *+/hIC1* to clarify the sole effect of *H19* overexpression. Here, 46 DEGs were identified (*Figure 6—figure supplement 2B*), which were enriched in vascular endothelial cell proliferation pathways (*Figure 6—figure supplement 2C*). Surprisingly, although physiologically similar to *+/hIC1*, the *ΔH19/hIC1* transcriptome was quite different from that of *+/hIC1*. Compared to wild-type, *ΔH19/hIC1* only had 23 DEGs (*Figure 6—figure supplement 2A*), in contrast to *+/hIC1* showing more than 200 DEGs compared to wild-type

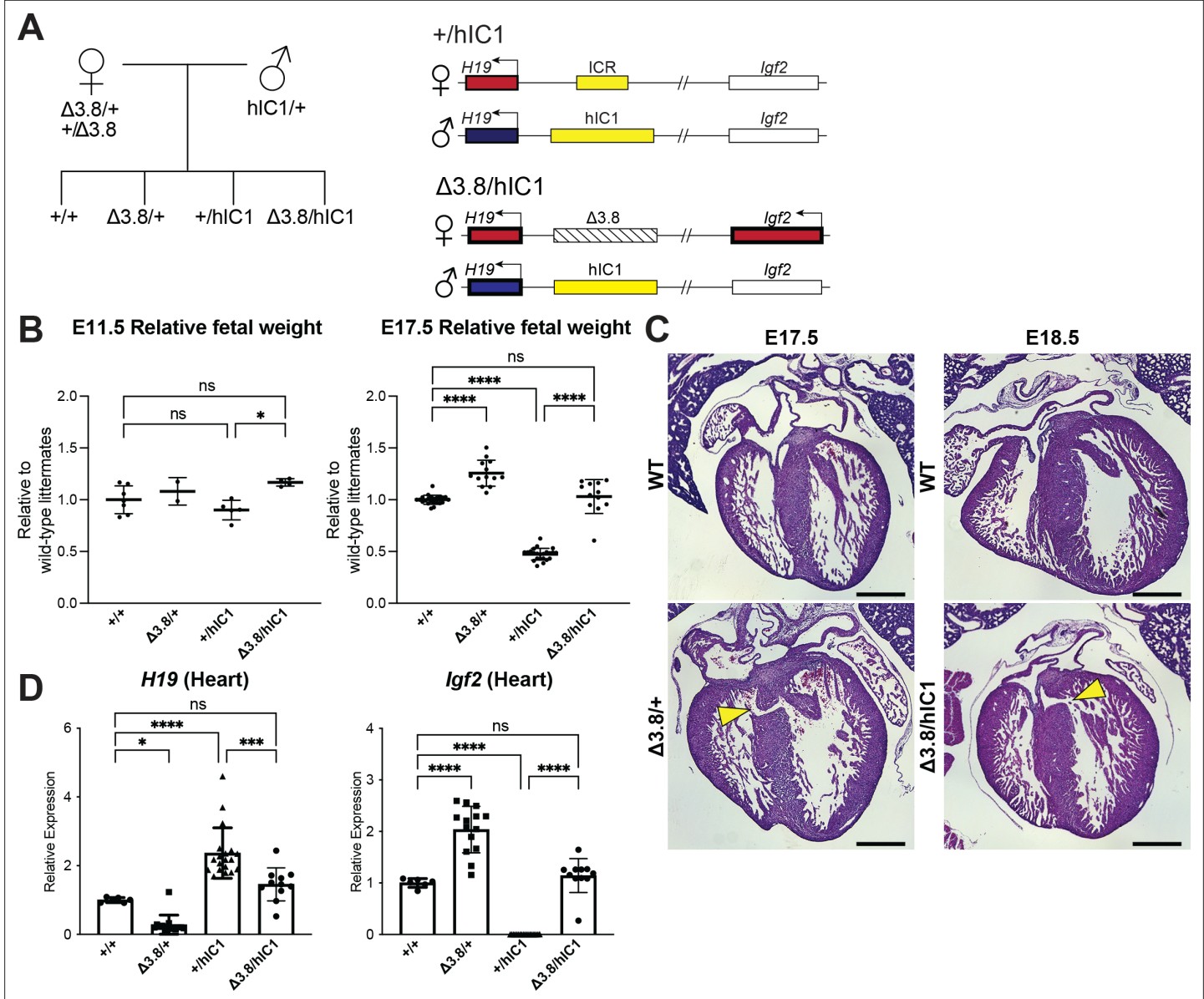

**Figure 5.** Restoring *H19* and *Igf2* expression utilizing maternal *H19/Igf2* imprinting control region (ICR) deletion. (**A**) A schematic representation of the offspring produced when △*3.8* heterozygous female and *hIC1/+* male mice are mated is depicted. △*3.8/hIC1* embryos are expected to show activation of maternal *Igf2* expression as well as paternal *H19* expression. (**B**) Relative fetal weights of wild-type, △*3.8/+*, *+/hIC1*, and △*3.8/hIC1* embryos at E11.5 and E17.5, normalized to the average body weight of wild-type littermates (mean ± SD). 2 E11.5 litters and 10 E17.5 litters are presented. (**C**) Representative cross-sections of E17.5 △*3.8/+* and E18.5 △*3.8/hIC1* embryonic hearts with ventricular septal defects (VSDs; yellow arrows), stained with hematoxylin and eosin. Sections from wild-type littermates are shown together for comparison. The E17.5 wild-type sample is male, and the rest are female. Scale bars = 500 μm. (**D**) Relative total expression of *H19* and *Igf2* in E17.5 wild-type, △*3.8/+*, *+/hIC1*, and △*3.8/hIC1* hearts (mean ± SD). (**B and D**) Each data point represents an individual conceptus from different litters. One-way ANOVA with Tukey's multiple comparisons test was used with *p<0.05, ***p<0.001, ****p<0.0001, and ns = not significant.

The online version of this article includes the following figure supplement(s) for figure 5:

**Figure supplement 1.** Supplementary data for rescue upon maternal *Δ3.8* transmission.

(*Figure 6B*, left). This result underscores the overwhelming effect of increased *H19* in transcriptomic regulation.

To identify the genes whose expression is solely affected by *H19*, we compared *+/hIC1*, *ΔH19/hIC1*, and *ΔH19/+* endothelial cells. Among 224 genes that are differentially expressed in *+/hIC1* endothelial cells relative to wild-type, 15 genes are also differentially expressed in *ΔH19/hIC1* samples,

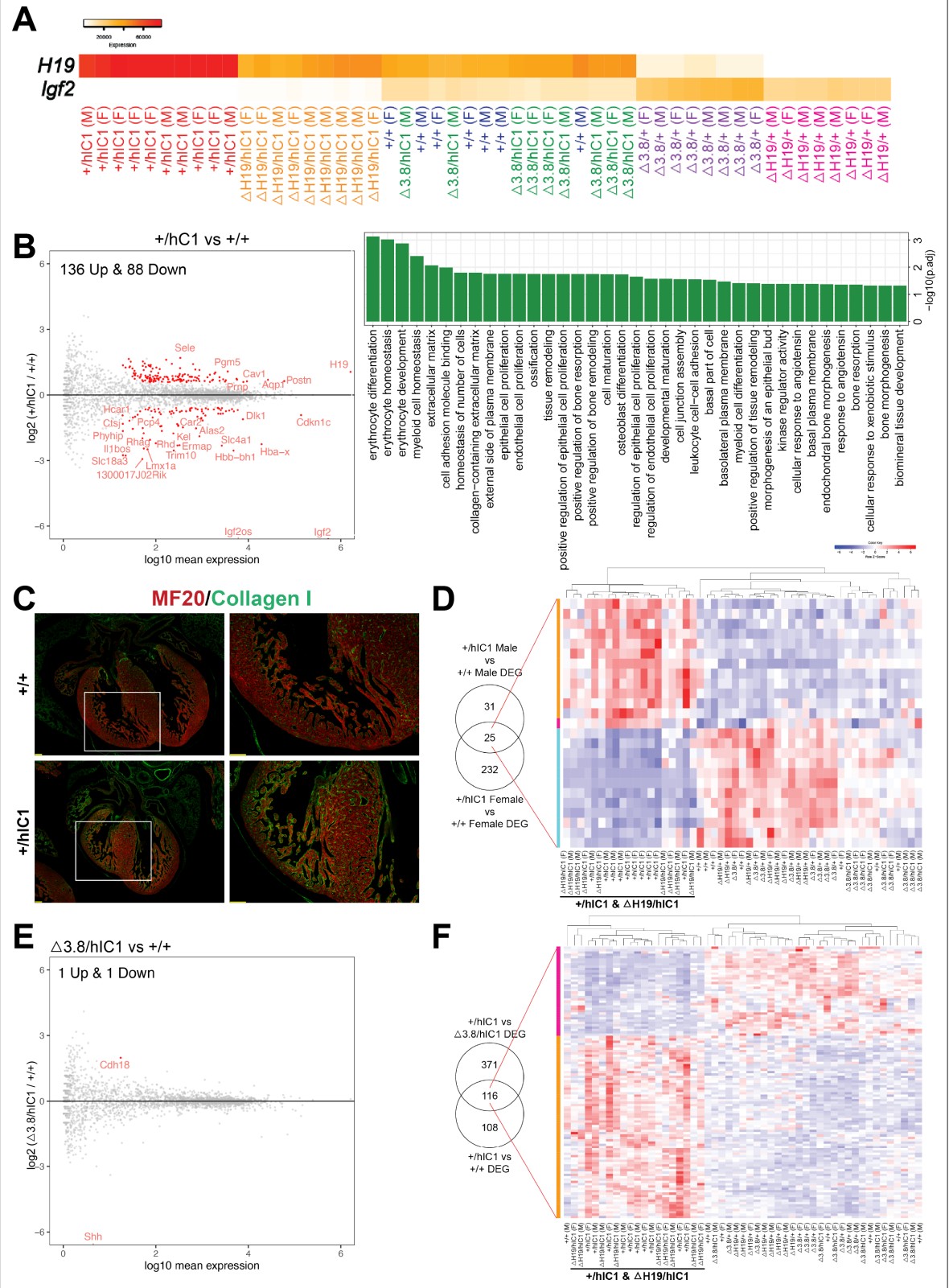

**Figure 6.** Transcriptomic analysis of E12.5 cardiac endothelial cells from wild-type and mutant embryos. (**A**) A gradient *H19* and *Igf2* expression levels are depicted in E12.5 wild-type, *+/hIC1*, Δ*H19/+*, Δ*H19/hIC1*, Δ*3.8/+*, and Δ*3.8/hIC1* cardiac endothelial cells. M: male and F: female. (**B**) Left: Comparison between *+/hIC1* and the wild-type samples with a volcano plot shows 224 differentially expressed genes (DEGs) between *+/hIC1* and the wild-type samples. Right: Gene ontology (GO) pathways that are enriched for the 224 DEGs. (**C**) Immunofluorescence staining for MF20 (red) and

*Figure 6 continued on next page*

*Figure 6 continued*

collagen (green) on E17.5 wild-type and *+/hIC1* hearts. Images on the right are enlarged from the boxed area of images on the left. Scale bars = 100 μm. (**D**) Expression pattern of 25 genes that are differentially expressed in both male and female *+/hIC1* samples is compared to wild-type. (**E**) A volcano plot represents two DEGs between *△3.8/hIC1* and the wild-type samples. (**F**) Expression pattern of 116 genes that are commonly differentially expressed in *+/hIC1* compared to the wild-type and *△3.8/hIC1* samples.

The online version of this article includes the following figure supplement(s) for figure 6:

**Figure supplement 1.** Supplementary data for differential gene expression of *+/hIC1* hearts.

**Figure supplement 2.** Further RNA-seq analysis involving *ΔH19/hIC1* endothelial cells.

**Figure supplement 3.** Enrichment tests for BRIDEAU_IMPRINTED_GENES using gene set enrichment analysis (GSEA).

**Figure supplement 4.** Further RNA-seq analysis utilizing *Δ3.8/+* and *Δ3.8/hIC1* endothelial cells.

**Figure supplement 5.** Enrichment tests for HALLMARK_HYPOXIA, GROSS_HYPOXIA_VIA_HIF1A_DN using gene set enrichment analysis (GSEA).

suggesting that these 15 genes were affected by the loss of *Igf2* rather than increased *H19*. Among the remaining 209 DEGs that are not altered in *ΔH19/hIC1* samples, only *Fgf10* and *H19* were also differentially expressed in *ΔH19/+* endothelial cells compared to wild-type. *Fgf10*, a key regulator of cardiac fibroblast development, mediates communication between cardiac progenitor cells and regulates cardiac myocyte proliferation (*Hubert et al., 2018*; *Vega-Hernández et al., 2011*). Consistently, *Fgf10* null mouse embryos showed abnormal cardiac morphology with reduced heart size and thinned ventricular wall (*Rochais et al., 2014*; *Vega-Hernández et al., 2011*). In our samples, *Fgf10* is upregulated when *H19* is deleted and downregulated upon *H19* overexpression, linking *H19* and *Fgf10* closely in the context of cardiac development. Additionally, gene set enrichment analysis revealed that the set of imprinted genes (BRIDEAU_IMPRINTED_GENES) including *Cdkn1c*, *Dlk1*, and *Gatm* was differentially enriched in *+/hIC1* samples (*Figure 6—figure supplement 3A*; *Mootha et al., 2003*; *Subramanian et al., 2005*). The same set of genes was also differentially enriched in *ΔH19/+* samples (*Figure 6—figure supplement 3B*), but not *ΔH19/hIC1* (*Figure 6—figure supplement 3C*), underscoring a role for *H19* as a master regulator of the imprinted gene network (*Gabory et al., 2010*).

We then analyzed *Δ3.8/hIC1* samples to identify cellular processes that are required to rescue the paternal *hIC1*-associated lethality, as *Δ3.8/hIC1* mice are fully viable with occasionally observed VSD. *Δ3.8/hIC1* had only two DEGs compared to wild-type (*Figure 6E*). However, the *Shh* expression was only detected in two wild-type samples, suggesting that *Shh* did not have significant affect in the development of our wild-type and *Δ3.8/hIC1* cardiac endothelial cells at this stage. In contrast, 487 genes were differentially expressed in *Δ3.8/hIC1* compared to *+/hIC1* (*Figure 6—figure supplement 4A*). Within these DEGs, we wanted to clarify the genes that were likely involved in restoring viability. The 487 DEGs between *+/hIC1* and *Δ3.8/hIC1* were compared to the DEGs between *+/hIC1* and wild-type to filter genes that are commonly affected in both comparisons (*Figure 6F*, left). GO analysis showed that the 116 overlapping genes are associated with endothelial/epithelial cell proliferation and remodeling (*Figure 6—figure supplement 4B*), emphasizing the importance of the proper regulation of these pathways in the rescued viability of *Δ3.8/hIC1* embryos. The expression pattern of these 116 genes clustered *+/hIC1* and *ΔH19/hIC1*, implicating these genes in the perinatal lethality characteristic of these two groups (*Figure 6F*, right).

## Discussion

In this study, we report that the overexpression of *H19* combined with *Igf2* depletion leads to severe morphological defects in the heart and placenta, which are likely to be involved with the perinatal lethality and restricted growth of SRS mouse models. Genetically correcting *H19* was not sufficient to fully rescue the developmental defects, indicating that the SRS-like phenotypes of paternal *hIC1* transmission are not solely attributable to *H19* overexpression. Unexpectedly, although moderately adjusting both *H19* and *Igf2* rescued the lethality, septal defects persisted in some of the *Δ3.8/hIC1* embryos, suggesting that cardiac development is extremely sensitive to the dosage of *H19* and *Igf2*. Our transcriptomic profiling of cardiac endothelial cells with various levels of *H19* and *Igf2* expression uncovers critical pathways driven by *H19* and *Igf2* that are important for cardiac structure formation. The result suggests that the regulation of ECM and proliferation of endothelial cells are tightly regulated by *H19* and *Igf2* and potentially responsible for the paternal *hIC1*-associated cardiac defects.

The function of *H19* in cardiac development is understudied even though the expression of *H19* is robust in the developing endocardium and epicardium throughout gestation (*García-Padilla et al., 2019*). Abnormal *H19/Igf2* expression in *+/hIC1* hearts disrupted AV cushion fusion, ventricular septation, and valve formation processes with variable penetrance. These events require properly established ECM, which accommodates endothelial-mesenchymal transition, cell proliferation, and cell-cell adhesion in developing hearts (*Kruithof et al., 2007*; *Sullivan and Black, 2013*; *von Gise and Pu, 2012*). In adult murine hearts, where *H19* has been well implicated in cardiac fibrosis and remodeling (*Greco et al., 2016*; *Hobuß et al., 2020*; *Lee et al., 2011*; *Wang et al., 2021*), *H19* overexpression led to increased ECM and fibrosis markers after myocardial injury, while deleting *H19* resulted in downregulated ECM genes (*Choong et al., 2019*). In our *+/hIC1* endothelial cells, the expression of several key ECM genes such as *Periostin* (*Postn*) (*Snider et al., 2009*; *Sullivan and Black, 2013*), *Col14a1*, and *Adamts17* (*Hubmacher and Apte, 2015*) was significantly upregulated compared to wild-type cells. Combined with increased collagen in E17.5 *+/hIC1* hearts (*Figure 6C*), we hypothesize that failing to establish proper ECM contributes significantly to the *+/hIC1*-associated cardiac defects during development. Additionally, regulators of cell proliferation, including *E2f5*, *Trabd2b*, and *Septin4*, and genes associated with inflammation such as *Sele* and *Cd200* were also differentially expressed in *+/hIC1* samples. Overall, this work provides some hints regarding the potential mechanism underlying how increased *H19* expression disrupts normal cardiac development.

In contrast to the less well understood role for *H19*, *Igf2* is the main growth factor in the developing ventricular wall. Similar to *H19*, *Igf2* is highly expressed in the developing cardiac endocardium and epicardium from early gestation (*Shen et al., 2015*) before ventricular septation is completed (*Savolainen et al., 2009*). Deletion of *Igf2* and its receptors caused decreased cardiomyocyte proliferation and ventricular hypoplasia, suggesting that *Igf2* is the major regulator of ventricular wall thickening (*Li et al., 2011*; *Shen et al., 2015*). Additionally, the IVS is comprised of both mesenchymal and muscular components (*Penny and Vick, 2011*; *Spicer et al., 2014*), and a reduction in cardiomyocyte proliferation can lead to VSD (*Snider and Conway, 2011*). Thus, the lack of *Igf2*, a growth promoter for cardiomyocytes, could have contributed to the septal defects observed in *+/hIC1* and *ΔH19/hIC1* hearts. However, the thinned myocardium in *Igf2* knockout mouse embryos was resolved by birth, resulting in normal cardiac morphology in neonates (*Shen et al., 2020*). In contrast, ventricular wall thinning and septal defects in *+/hIC1* and *ΔH19/hIC1* hearts were aggravated toward the end of gestation, indicating that these phenotypes are not exclusively attributable to the loss of *Igf2* expression.

Recovery of *Igf2* expression in *Δ3.8/hIC1* hearts rescued ventricular hypoplasia, consistent with previous findings (*Li et al., 2011*; *Shen et al., 2015*). In contrast, septal defects persist in some *Δ3.8/+* and *Δ3.8/hIC1* embryos. Because *Igf2* expression varied substantially among *Δ3.8/+* and *Δ3.8/hIC1* hearts (*Figure 5D*), we hypothesize that the varying penetrance of septal defects in these hearts reflects the range of *H19/Igf2* levels. The only upregulated gene in *Δ3.8/hIC1* endothelial cells compared to wild-type was *Cadherin 18 (Cdh18)* (*Figure 6E*), which was also upregulated in *Δ3.8/+* samples compared to wild-type (*Figure 6—figure supplement 4C*) and clinically reported to be mutated in congenital heart defects (CHDs) including VSD (*Chen et al., 2018*; *Soemedi et al., 2012*). Although we and others showed that restoring *Igf2* successfully rescues the growth restriction in SRS-like mouse models (*Han et al., 2010*; *Liao et al., 2021*), septal defects caused by *H19* and *Igf2* dysregulation are reported for the first time in this study. Our data provide evidence that ventricular septation may be regulated separately from the ventricular wall thickening and that both events are extremely sensitive to the level of *H19* and *Igf2* expression. In humans, VSDs were found in SRS patients with IC1 hypomethylation (*Ghanim et al., 2013*) and a patient carrying chromosomal gain of chr11p15 (*Serra et al., 2012*). Thus, the VSD observed in our mouse models nicely models the SRS-associated CHD in human patients. It should be noted, however, that mice are often more susceptible to VSDs than humans. Mice with VSDs show a higher neonatal mortality rate, possibly due to the higher heart rates and relatively larger size of VSD lesions (*Snider and Conway, 2011*). Additionally, cardiac defects that are lethal in mice can lead to spontaneous miscarriages in humans due to longer human gestation, making it difficult to observe human infants with similar defects.

Linked through fetoplacental blood circulation, the heart and placenta are closely connected under the cardiac-placental axis, which is crucial for fetal growth and viability (*Barak et al., 2019*; *Maslen, 2018*). Both organs are responsible for supplying nutrients and oxygen for developing fetuses, and

placental and cardiac defects are often coupled in mouse models (*Perez-Garcia et al., 2018*) and humans (*Matthiesen et al., 2016*; *Rychik et al., 2018*). Placental maintenance of a normoxic fetal environment is vital for cardiac morphogenesis, and hypoxia can lead to severe defects emerging especially in cushions and septa (*Dor et al., 2001*). Epicardial *Igf2* expression in developing ventricles is induced by a normoxic environment that is dependent on the placental function (*Shen et al., 2015*), while *H19* is upregulated under hypoxic conditions in mouse cardiomyocytes (*Choong et al., 2019*). This suggests that *H19* and *Igf2* are important mediators of the interaction between heart and placenta. Consistent with previous reports, gene sets of which expression is altered under hypoxic conditions (HALLMARK_HYPOXIA, GROSS_HYPOXIA_VIA_HIF1A_DN) are differentially enriched in our *+/hIC1* endothelial cells compared to wild-type (*Figure 6—figure supplement 5*). As the heart and placenta are responsible for meeting the fetal demand for nutrients and oxygen, the malfunction of these two organs would severely constrain embryonic growth. Therefore, the precise role of *H19* and *Igf2* in cardiac-placental communication needs to be clarified to understand how the SRS-related growth restriction is induced by IC1 hypomethylation.

The labyrinth, where *+/hIC1* placentas show abnormal vasculature morphology and thrombosis, serves as a prime location for maternal-fetal blood exchange (*Woods et al., 2018*). Placental thrombosis can be caused by defective labyrinth integrity, and diminished labyrinth function could limit the nutritional and oxygen supply for a fetus, which can, in turn, lead to hypoxia and growth restriction. Both *H19* and *Igf2* are highly expressed in fetoplacental endothelial cells in the labyrinth (*Aykroyd et al., 2022*; *Sandovici et al., 2022*). Genetically depleting *Igf2* expression in the epiblast lineage led to decreased labyrinth size and caused the formation of thrombi in the labyrinth, although the lesions were smaller in size than those observed in *+/hIC1* placentas (*Sandovici et al., 2022*). Depletion of the placental-specific *Igf2* transcript in mice resulted in a smaller labyrinth and fetal growth restriction (*Constância et al., 2002*; *Sibley et al., 2004*), although these mice showed an increased fetal to placental weight ratio, which was decreased in *+/hIC1* mice. This contrasting result could be explained by the effect of the increased *H19* expression in *+/hIC1* mice. *H19* regulates vascular endothelial growth factor in human endothelial cells in vitro (*Conigliaro et al., 2015*), which is involved in placental angiogenesis, specifically for the branching of fetoplacental vessels beginning at midgestation (*Woods et al., 2018*). Additionally, *H19* is highly expressed in trophoblasts (*Marsh and Blelloch, 2020*; *Poirier et al., 1991*), and disrupted trophoblast development leads to defective vascular branching in the labyrinth and restricted fetal growth (*Ueno et al., 2013*). Thus, it is possible that the morphological anomalies observed in the *+/hIC1* placenta are exaggerated by abnormal *H19* expression in trophoblasts. Transcriptomic analysis of fetoplacental endothelial cells from our mouse models would help us understand the role of *H19/Igf2* in placental development. Moreover, generating a tissue-specific *hIC1* mouse model, if possible, would allow us to determine the causal relationship between cardiac and placental phenotypes.

In summary, we provide evidence that the proper dosage of *H19* and *Igf2* is essential for normal cardiac and placental development. Investigation of the role of *H19* and *Igf2* in the cardiac-placental axis will enable a better understanding of how paternal *hIC1* transmission leads to the SRS-like growth restriction and perinatal lethality. As many patients with SRS exhibit DNA methylation mosaicism, the distribution of the epimutation, which reflects the severity of patient symptoms, varies. This work provides insight into identifying organs that are most sensitive to *H19* and *Igf2* dysregulation, which would allow us to develop early intervention methods for critical SRS pathologies with such variabilities.

## Materials and methods

### Animal studies

All studies were approved by the Institutional Animal Care and Use Committee at the University of Pennsylvania. *hIC1* (*Hur et al., 2016*) and *Δ3.8* (*Thorvaldsen et al., 2002*; *Thorvaldsen et al., 2006*) mouse models were previously described. Timed breeding was performed as previously described (*SanMiguel et al., 2018*). Vaginal sperm plugs were checked to calculate the age, and the day of the plug was marked as E0.5. Visual staging confirmed the embryonic days at the time of dissection. All mice were maintained on C57BL/6 background for more than 10 generations if not noted otherwise.

## Generation of ΔH19 allele

Two pairs of gRNA targeting the *H19* gene are listed in *Supplementary file 1*. gRNA was prepared following a protocol from *Yang et al., 2014* with modifications. px335 plasmid (Addgene, Watertown, MA, USA) was PCR amplified using Phusion high-fidelity DNA polymerase (New England Biolabs, Ipswich, MA, USA) and the primer set listed in *Supplementary file 1*. ~117 bp PCR product was gel-purified using the Gel Extraction kit (Qiagen, Hilden, Germany) according to the manufacturer's instruction. Using the gel-purified product as template, in vitro transcription of gRNA was setup using T7 High Yield RNA Synthesis kit (New England Biolabs) according to manufacturer's instructions. Transcribed gRNA was purified using the MEGAclear kit (Life Technologies, Carlsbad, CA, USA) according to manufacturer's instructions. 50 ng/μl left and right single guide RNA (sgRNA) together with 100 ng/μl Cas9 mRNA was injected per zygote stage embryo by the Transgenic and Chimeric Mouse Facility at the University of Pennsylvania. The targeted allele was validated using Southern blot as previously described (*Thorvaldsen et al., 1998*) and PCR and sequencing across junctions of *ΔH19* alleles. Obtained chimeras and germ line transmission animals were PCR-genotyped for the *ΔH19* allele using primers (*Supplementary file 1*).

## Genotyping

Mouse genomic DNA for PCR genotyping was isolated from each animal as previously described (*SanMiguel et al., 2018*). Primers used for sex genotyping and genotyping of *hIC1*, *Δ*H19, and *Δ*3.8 alleles are listed in *Supplementary file 1*. For all genotypes, the maternal allele is listed first and the paternal allele second.

## Gene expression analysis

Mouse tissues were ground using pestles, syringes, and needles in either TRIzol (Thermo Fisher scientific, Waltham, MA, USA) or RLP buffer included in RNeasy Mini kit (Qiagen, Hilden, Germany). RNA was isolated according to the manufacturer's instructions. cDNA synthesis, quantitative reverse transcription (qRT)-PCR, and allele-specific expression analysis was performed as previously described (*Hur et al., 2016*). Primers and PCR conditions are listed in *Supplementary file 1*. Total expression levels of *Mir675-3p* and *Mir675-5p* were determined relative to snoRNA202 by using a separate RT kit (TaqMan MicroRNA Reverse Transcription Kit, Thermo Fisher Scientific), qRT-PCR primers (Assay Id 001232, 001941, 001940, Thermo Fisher Scientific), and a PCR master mix (TaqMan Universal PCR Master Mix, catalog number 4304437, Thermo Fisher Scientific) according to manufacturer's protocol.

## Histological analysis

Mouse heart and placenta samples were collected in cold PBS, fixed overnight in 4% paraformaldehyde or 10% phosphate-buffered formalin and processed through ethanol dehydration. Tissues were paraffin-embedded and sectioned for further staining analysis. Hematoxylin and eosin staining was performed using a standard protocol. Immunohistochemistry for MF20 (Developmental Studies Hybridoma Bank, Iowa City, IA, USA) and Collagen I (cat# ab34710, Abcam, Cambridge, UK) was performed with primary antibody incubations overnight at 4°C. Prior to antibody incubation, antigen retrieval with citrate buffer was performed, followed by a 1 hr block in 10% normal serum. DAPI (cat#32670–5 MG-F, Sigma-Aldrich, St. Louis, MO, USA) was used as a counter stain, and slides were mounted with VECTASHILD (Vector, Burlingame, CA, USA). Images were taken on a Leica DMi8S widefield microscope. Placental CD34 staining and PAS staining was previously described (*Vrooman et al., 2020*). The thickness of ventricular wall was measured on hematoxylin-eosin-stained heart sections using Adobe Photoshop. A minimum of three distinct sections were quantified for each mouse of each genotype in a blinded manner. Placental Jz/Lb ratio was measured using FIJI (ImageJ v2.0.0, *Schindelin et al., 2012*) in a blinded manner. CD34 stained placental sections were digitally scanned using Aperio VERSA 200 platform in Comparative Pathology Core at School of Veterinary Medicine at the University of Pennsylvania. Images were analyzed via Aperio Microvessel Analysis algorithm as previously described (*Vrooman et al., 2020*).

## RNA sequencing library preparation and analysis

E12.5 hearts were lysed with Collagenase (Sigma-Aldrich), Dispase II (Sigma-Aldrich), and DNase I (Sigma-Aldrich). Cardiac endothelial cells were collected using MACS CD31 microbeads (Miltenyi

Biotec, Bergisch Gladbach, Germany), and RNA was isolated using RNeasy Micro kit (Qiagen). After confirming RNA integrity using Bioanalyzer (Agilent Technologies, Santa Clara, CA, USA), mRNA library was generated from 25 ng RNA using NEBNext Poly(A) mRNA Magnetic Isolation Module and Ultra II RNA Library Prep Kit (New England Biolabs). Library quality was assessed by Bioanalyzer (Agilent Technologies) and TapeStation (Agilent Technologies). Sequencing was performed on NovaSeq 6000 (Illumina, San Diego, CA, USA). Quality of raw fastq reads was assessed using FastQC version 0.11.5 (*Andrews et al., 2015*). Reads were aligned to the GRCm38/mm10 reference using STAR version 2.4.0i with default parameters and maximum fragment size of 2000 bp (*Dobin et al., 2013*). Properly paired primary alignments were retained for downstream analysis using Samtools version 1.9. Count matrices were generated using FeatureCounts version 1.6.2 against RefSeq gene annotation and read into DESeq2 (*Love et al., 2014*) to perform normalization and statistical analysis.

## Statistical analysis

Differences between two groups were evaluated using Student's t-test. For three or more groups, ordinary one-way ANOVA, followed up with Tukey's multiple comparisons test, was used. Two-sided Fisher's exact test was used to compare the occurrence of the thrombi in the wild-type and *+/hIC1* placentas. All analyses were performed using GraphPad Prism software. $*p<0.05$, $**p<0.01$, $***p<0.001$, $****p<0.0001$, and n.s.=not significant.

## Acknowledgements

Authors would like to express gratitude to Christopher Krapp, Lisa Vrooman, Joel Rurik, Olga Smirnova, Jonathan Schug, Klaus Kaestner, and Colin Conine for their guidance on this study. This work was supported by National Institutes of Health grant GM-051279–28 and R35HL140018-05, National Institute of Arthritis and Musculoskeletal and Skin Diseases grant 5T32AR053461.

## Additional information

### Funding

| Funder | Grant reference number | Author |
|---|---|---|
| National Institutes of Health | GM-051279-28 | Marisa S Bartolomei |
| National Institutes of Health | R35HL140018-05 | Jonathan A Epstein |
| National Institute of Arthritis and Musculoskeletal and Skin Diseases | 5T32AR053461 | Jonathan A Epstein |

The funders had no role in study design, data collection and interpretation, or the decision to submit the work for publication.

### Author contributions

Suhee Chang, Conceptualization, Resources, Data curation, Software, Formal analysis, Validation, Investigation, Visualization, Methodology, Writing – original draft, Project administration, Writing – review and editing; Diana Fulmer, Resources, Data curation, Funding acquisition, Validation, Investigation, Visualization, Methodology, Project administration, Writing – review and editing; Stella K Hur, Conceptualization, Resources, Data curation, Investigation, Methodology; Joanne L Thorvaldsen, Conceptualization, Resources, Data curation, Validation, Investigation, Visualization, Methodology, Project administration, Writing – review and editing; Li Li, Resources, Validation, Investigation; Yemin Lan, Data curation, Software, Formal analysis, Writing – review and editing; Eric A Rhon-Calderon, Resources, Data curation, Visualization, Methodology; Nicolae Adrian Leu, Validation, Investigation, Writing – review and editing; Xiaowen Chen, Resources, Methodology, Writing – review and editing; Jonathan A Epstein, Resources, Supervision, Funding acquisition, Project administration, Writing – review and editing; Marisa S Bartolomei, Conceptualization, Resources, Data curation, Supervision,

Funding acquisition, Visualization, Methodology, Writing – original draft, Project administration, Writing – review and editing

#### Author ORCIDs
Suhee Chang http://orcid.org/0000-0002-0514-3516
Diana Fulmer http://orcid.org/0000-0001-7675-9951
Jonathan A Epstein http://orcid.org/0000-0001-8637-4465
Marisa S Bartolomei http://orcid.org/0000-0001-9410-5222

#### Ethics
All of the animals were handled according to approved institutional animal care and use committee (IACUC) protocols (#804211) of the University of Pennsylvania. All the animals were euthanized by carbon dioxide inhalation as recommended by National Institutes of Health. As a secondary method of euthanasia, either decapitation or cervical dislocation was performed, and every effort was made to minimize suffering.

#### Decision letter and Author response
Decision letter https://doi.org/10.7554/eLife.78754.sa1
Author response https://doi.org/10.7554/eLife.78754.sa2

## Additional files

#### Supplementary files
• Supplementary file 1. Primers and PCR conditions utilized in this study.
• MDAR checklist

#### Data availability
Sequencing data have been deposited in GEO database under the accession number GSE199377.

The following dataset was generated:

| Author(s) | Year | Dataset title | Dataset URL | Database and Identifier |
|---|---|---|---|---|
| Chang S, Fulmer D, Hur SK, Thorvaldsen JL, Li L, Lan Y, Rhon-Calderon EA, Chen X, Epstein JA, Bartolomei MS | 2022 | Dysregulated H19/Igf2 expression disrupts cardiac-placental axis during development of Silver Russell Syndrome-like mouse models | http://www.ncbi.nlm.nih.gov/geo/query/acc.cgi?acc=GSE199377 | NCBI Gene Expression Omnibus, GSE199377 |

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
