## [Editor Report]

This important study took advantage of multiple mouse models with varying levels of H19 and Igf2 expression to dissect the role of H19 and Igf2 in cardiac and placental development. This work links the heart and placenta through regulation by H19 and Igf2, demonstrating that an accurate dosage of both H19 and Igf2 is critical for normal embryonic development, especially related to the cardiac-placental axis. The topic is of significance, and the data are of high quality and convincing.

---

## [Decision Letter]

**Decision letter after peer review:**

Thank you for submitting your article "Dysregulated *H19/Igf2* expression disrupts cardiac-placental axis during development of Silver Russell Syndrome-like mouse models" for consideration by *eLife*. Your article has been reviewed by 3 peer reviewers, one of whom is a member of our Board of Reviewing Editors, and the evaluation has been overseen by Marianne Bronner as the Senior Editor. The following individual involved in the review of your submission has agreed to reveal their identity: Amanda N Sferruzzi-Perri (Reviewer #3).

Essential revisions:

1) A tetraploid aggregation experiment in which the placental trophoblast lineage is rescued in at least one of the models, ideally the +/hIC1 mice.

2) The molecular interactions between H19 and Igf2 should be explored if possible.

*Reviewer #1 (Recommendations for the authors):*

Work on the genetic dissection of effects of either individual or combined dysregulation of H19 and Igf2 on cardiac and placental development was well done. However, the molecular interactions between H19 and Igf2 were not explored. With multiple genetic models in hand, it would be interesting to examine changes in miRNAs and H19 lncRNAs as well as Igf2 expression levels to determine whether the existing notions (miR-645 produced from H19 lncRNA targets Igf2) are correct or not; it can also lead to novel mechanistic insight into interactions between H19 and Igf2.

*Reviewer #2 (Recommendations for the authors):*

This is a well-written and thoroughly conducted study by a world-leading team that has specialized in the fine analysis of Igf2 and H19 regulation and function.

In addition to the data provided, the study would be immensely strengthened by a tetraploid aggregation approach in which the placental trophoblast lineage is rescued in at least one of the models, ideally the +/hIC1 mice. This experiment would identify whether the hemorrhagic areas in the placenta are caused by endothelial cell dysfunction, or conversely whether the heart defects are fully or partially caused by trophoblast dysfunction.

In addition to the comments in the public review, the use of bean plots is not always the best display. In particular, the diagram on Mendelian ratios is clear where the 'bean' reaches negative values. A simple scatter plot or box plot display will be fine.

Placental histology must be improved, and robust vascular staining images supplied.

Figure 4D bottom right, the heart is sectioned at an angle, please replace this specimen with a more suitable example.

*Reviewer #3 (Recommendations for the authors):*

1. Line 69-72; 'Although the epigenetic defects and growth restriction of these mice model SRS, the epigenetic defects in human are mosaic (Soellner et al., 2019), with a subset of cells showing normal methylation patterns, providing a potential explanation for why SRS individuals are viable.' Reword to increase clarity.

2. Line 91-92: 'Normalization of H19 rescued neither the lethality nor growth restriction, although we observed a minimal rescue of the earlier growth and resorption frequency.' Reword to increase clarity.

3. L119 milk spots where? In their abdomen?

4. It is excellent that the authors used litter means for their statistical analyses. This robust method should be applied to Figure 3B, which should represent litter means, not individual fetuses.

5. Figure 3C and H: what do the violin plots represent? Confidence intervals? What is the sex of the fetuses shown in 3G? please state in legend. Part of Figure 3H x-axis label cant be seen. Was microvessel density also only reduced in +/hIC1 males?

6. When do the defects in the placenta microvasculature appear relative to the heart defects in +/hIC1? Were growth and heart defects more common/severe in males?

7. All figures, including supplementary should have the y-axes starting at 0 – which is better for identifying the magnitude of the change. They should all include information on the number of litters used and employ litter averages for statistics (eg Supp Figure 3D is unclear).

8. Text stating Data not shown: Please show all data or do not describe it.

9. Using in silico/bioinformatic analysis, are the authors able to estimate what proportion of the DEGs may be directly (vs indirectly) modified by the H19 ncRNA in the relevant mutants?

10. I realise it is out of the scope of the paper, but it would have been amazing to have undertaken similar transcriptomic analyses of the placenta at E12.5 – to assess how the placenta and heart may similarly or differentially respond to the H19/Igf2 gene alterations. Moreover, undertaking placental or cardiac-specific gene manipulations would inform further on the nature of the plac-heart axis and how it may be mediated through changes in the H19/Igf2 imprinted locus. Descriptions of these future studies should be included in the discussion.

---

## [Author Response]

Essential revisions:1) A tetraploid aggregation experiment in which the placental trophoblast lineage is rescued in at least one of the models, ideally the +/hIC1 mice.

See Reviewer 2, point 1.

2) The molecular interactions between H19 and Igf2 should be explored if possible.

See Reviewer 1, point 1

Reviewer #1 (Recommendations for the authors):Work on the genetic dissection of effects of either individual or combined dysregulation of H19 and Igf2 on cardiac and placental development was well done. However, the molecular interactions between H19 and Igf2 were not explored. With multiple genetic models in hand, it would be interesting to examine changes in miRNAs and H19 lncRNAs as well as Igf2 expression levels to determine whether the existing notions (miR-645 produced from H19 lncRNA targets Igf2) are correct or not; it can also lead to novel mechanistic insight into interactions between H19 and Igf2.

In order to determine if the level of *Mir675* is correlated with changes in *H19* and *Igf2* expression*,* we examined the *Mir675* expression in *+/hIC1* placenta at E15.5 (Figure 3—figure supplement 1E). Although increased *H19* expression together with *Igf2* depletion was present in E15.5 +/hIC1 placentas (Hur et al., 2016), *Mir675* was not significantly increased in *+/hIC1* placentas compared to the wild-type. Thus, the placental phenotypes observed in *+/hIC1* mice are solely attributable to the increased *H19* lncRNA, irrespective of *Mir675*. This information was added to lines 173-183. Associated methods were also added to lines 446-450.

Reviewer #2 (Recommendations for the authors):1. In addition to the data provided, the study would be immensely strengthened by a tetraploid aggregation approach in which the placental trophoblast lineage is rescued in at least one of the models, ideally the +/hIC1 mice. This experiment would identify whether the hemorrhagic areas in the placenta are caused by endothelial cell dysfunction, or conversely whether the heart defects are fully or partially caused by trophoblast dysfunction.

As requested, we performed tetraploid aggregation experiments to test if wild-type placental trophoblast lineage could rescue the cardiac defects of *+/hIC1* embryos. The experimental method was described further in the figure A below and ‘Materials and methods’ below. Briefly, tetraploid (4N) cells, that were derived from wild-type 2-cell embryos, were aggregated with diploid (2N) embryos from mating between wild-type females and *hIC1/+* males (Tanaka et al., 2009). The *hIC1/+* males were homozygous for GFP inserted at *Oct4* locus (Lengner et al., 2007). Therefore, the absence of GFP in the extraembryonic gDNA would indicate that the extraembryonic tissue originated from the 4N cells, which did not contain GFP. Aggregated blastocysts were transferred into recipient females, and embryos and placentas were weighed and collected at E15.5 and E17.5 for further analysis.

It is known that in tetraploid aggregation, 2N cells are capable of contributing to both embryonic and extraembryonic tissues without restriction, although contribution from 4N cells is restricted to extraembryonic tissues (Tanaka et al., 2009). Therefore, our aggregated placenta will be chimeric for wild-type and *+/hIC1* cells in the best case. As expected, no placenta was fully derived from the wild-type 4N cells. In order to estimate the contribution of 4N cells, we performed qPCR analysis of hIC1 and/or GFP in embryonic and placental gDNA. For all experiments (at least 3 attempts at E17.5 and 1 attempt at E15.5), hIC1 and/or GFP gDNA was detected in both the embryo and placenta, indicating that 2N embryonic cells contributed to the formation of extraembryonic tissue.

At E17.5, we recovered four *+/hIC1* embryos with placenta containing 4N wild-type cells. In all four E17.5 embryos, the level of hIC1 was higher in embryonic gDNA compared to placental gDNA, indicating that 4N wild-type cells are contributing to these placentas (Author response image 1, left). In samples containing a GFP allele in 2N cells, we measured the level of GFP in embryonic and placental gDNA, as only in 2N cells but not in 4N cells had GFP. In these E17.5 recovered embryos, GFP-positive 2N cells always contributed to the placenta (Author response image 1, right). We did not observe rescue in fetal weight (Author response image 1) or cardiac and placental morphologies in the *+/hIC1* embryos, irrespective of the level of 4N wild-type cells in the placenta. Large thrombi were observed in placentas of *+/hIC1* embryos that were aggregated with 4N wild-type cells (Author response image 1). Such lesions were not found in placentas of wild-type embryos from the same aggregation (Author response image 1). Thus, the aggregated 4N wild-type cells did not rescue the placental morphology, even when the contribution of 2N *+/hIC1* cells to placenta was extremely low. As the placental defects were not rescued, it would be impossible to conclude whether or not placental dysfunction is causing the *+/hIC1-*associated cardiac defects utilizing these embryos. The varying contribution of 2N cells to placenta was also observed at E15.5. This was confirmed through measuring levels of hIC1 and GFP in embryonic versus placental gDNA (Author response image 1). Regardless of the level of contribution of wild-type 4N cells to the placenta, the embryonic and placental undergrowth as well as cardiac defects persisted in these embryos (Author response image 1).

We also aggregated 2N wild-type cells with 4N *+/hIC1* cells to test if *+/hIC1* placenta is sufficient to cause cardiac defects in embryos. However, out of 20 embryos we recovered at E17.5, only one embryo was found to have *+/hIC1* cells in its placenta, and the level of hIC1 in its placental gDNA was less than 7.5% of the hIC1 level in normal *+/hIC1* placentas. Therefore, this embryo was not suitable to test the effect of *+/hIC1* placentas on the cardiac development of the embryo.

In conclusion, we were fortunate to have a colleague who could perform the requested aggregation experiments and were quite hopeful regarding this line of experimentation. Nevertheless, although we spent the revision period addressing the reviewer request, we do not feel confident in stating whether the phenotypes originate from the heart (endothelial cells) or placenta (trophoblast cells). Optimally, an experiment in which the embryo or placenta were entirely derived with the hIC1 allele would be the best test see text at end of discussion and Review 3 response (Point 10). However, because the allele was quite complicated to obtain and transmit due to the lethality of the paternal-specific insertion, this experiment is not possible.

**Author response image 1. sa2fig1:** Tetraploid aggregation experiment. (**A**) Schematic description of tetraploid aggregation experiment. (**B**) Level of hIC1 and GFP in embryonic and placental gDNA from hIC1 aggregated samples. (**C**) Embryonic and placental weight of E17.5 aggregated samples. (**D**) Representative cross-sections of E17.5 4N wild-type placentas with *+/hIC1* embryos stained with hematoxylin and eosin. Black arrowhead indicates a large thrombus formed in the *+/hIC1* labyrinth. (**E**) Representative cross-sections of E17.5 4N wild-type placentas with *+/hIC1* embryos stained with hematoxylin and eosin. (**F**) (**G**) (**D, E**) Scale bars = 1mm. (**B, C, G**) Each data point represents an individual conceptus from different aggregated litters. Statistics used are (**B, C, G**) multiple unpaired t-test. *P < 0.05, **P < 0.01, ***P < 0.001, ****P < 0.0001, ns = not significant.

“Materials and methods:

The embryos used for this experiment were collected from 4-6 week old C57BL/6 females. The females were superovulated using 5 IU of pregnant mare serum gonadotropin (PMSG) followed 48 hours later by 5 IU of human chorionic gonadotropin (hCG) after which they were mated to C57Bl6 and hlC1studs. For latter experiments, the *hIC1/+* stud males were also homozygous for a GFP knock-in allele (Lengner et al., 2007). All the hormones were administrated via intraperitoneal injection (IP).

The 2-cell embryos used for the aggregation experiment were collected from the oviduct about 42 hours post hCG. To generate the tetraploid embryos, 2-cell embryo blastomeres were fused together by using a CF-150/B cell fusing apparatus (voltage: DC 9; mico second 40; pulse 1). Only the embryos in which the blastomeres successfully fused were considered tetraploid embryos and were aggregated with the diploid embryos. The embryos were aggregated at the 2 and 4 cell stage. The zona pellucida was removed from both the diploid and tetraploid embryos with acidic tyrodes solution (EmbryoMax, cat# MR-004-D) and after removal, one diploid and one tetraploid embryo were placed together in a small dimple made in the plastic dish. The dimples were made using the aggregation needles (type:DN-09). About 20 aggregated embryo pairs were cultured per culture drop. All the aggregates were cultured until the blastocyst stage in a 5% O2, 5% CO2 incubator using KSOM media (Millipore, MR-202P-5F). The KSOM culture drops were covered with mineral oil (Millipore, ES-005-C) to prevent evaporation. Only the successfully aggregated embryos were transferred into Swiss Webster pseudo-pregnant recipient females which were synchronized by using Swiss Webster vasectomized males.

Embryo collection, embryo fusion and embryo transfers were performed at room temperature in HEPES-buffered CZB medium (PVP 0.1% w/v; albumin-free). We weighed embryos and placentas and collected heart and placenta for histology. We collected embryonic tissues and placenta for genotyping.”

2. The use of bean plots is not always the best display. In particular, the diagram on Mendelian ratios is clear where the 'bean' reaches negative values. A simple scatter plot or box plot display will be fine.

As recommended, we have replaced all of the bean/violin plots presented in (Figure 2B, 3C, 3H and 4B; Figure 4—figure supplement 2A and Figure 5—figure supplement 1C) with scatter plots.

3. Figure 4D bottom right, the heart is sectioned at an angle, please replace this specimen with a more suitable example.

As requested, we have replaced the angled E17.5 *dH19/hIC1* heart image in Figure 4D with a more suitable example.

Reviewer #3 (Recommendations for the authors):1. Line 69-72; 'Although the epigenetic defects and growth restriction of these mice model SRS, the epigenetic defects in human are mosaic (Soellner et al., 2019), with a subset of cells showing normal methylation patterns, providing a potential explanation for why SRS individuals are viable.' Reword to increase clarity.

We have reworded Lines 69-72 to: ‘Although the epigenetic defects and growth restriction of these mice nicely model SRS symptoms, many SRS individuals are viable. A potential explanation for such discrepancy between human and mouse could reflect the mosaic nature of epigenetic defects in human (Soellner et al., 2019), with a subset of cells showing normal methylation patterns.’ (see Lines 71-75).

2. Line 91-92: 'Normalization of H19 rescued neither the lethality nor growth restriction, although we observed a minimal rescue of the earlier growth and resorption frequency.' Reword to increase clarity.

We have reworded Lines 91-92 to:

‘Deletion of *H19* from the maternal allele, thereby reducing *H19* levels, failed to rescue completely the lethality and growth restriction associated with the paternal inheritance of hIC1. A minimal rescue of the earlier growth and resorption frequency was, however, observed with normalized *H19* expression.’ (see lines 94-96).

3. L119 milk spots where? In their abdomen?

To Line 119, location of milk spots has been added: ‘….milk spots were found in their abdomen,…’ (see line 123).

4. It is excellent that the authors used litter means for their statistical analyses. This robust method should be applied to Figure 3B, which should represent litter means, not individual fetuses.

The graph in Figure 3B was edited and now represents litter means. The following statement was added to the figure legend: ‘Each data point represents an average F/P ratio of each genotype from one litter. 13 litters are presented.’

5. Figure 3C and H: what do the violin plots represent? Confidence intervals? What is the sex of the fetuses shown in 3G? please state in legend. Part of Figure 3H x-axis label cant be seen. Was microvessel density also only reduced in +/hIC1 males?

We have replaced the violin plots in Figure 3C and H with simple scatter plots presenting mean with SD (also recommended by reviewer #2, Point 9), to be consistent with the presentation of other plots including Figure 3A, 3B and 3D. The distribution of individual samples can now be observed. In Figure 3G, the wild-type fetus is female and *+/hIC1* fetus is male. This information was added to the figure legend. The decrease in microvessel density was statistically significant only in *+/hIC1* male placentas. Still, *+/hIC1* female placentas showed a trend to reduced microvessel density compared to the wild-type, although to less extent and not statistically significant. We have also separated the wild-type male and female samples in Figure 3H to make the sex specific comparisons of microvessel density (Figure 3H; also see lines 162-164).

6. When do the defects in the placenta microvasculature appear relative to the heart defects in +/hIC1? Were growth and heart defects more common/severe in males?

The placental microvasculature defects were observed later than cardiac defects. At E12.5, which is the earliest stage when the cardiac defects were observed, CD34 staining showed that *+/hIC1* placentas do not have a significant difference in microvasculature structures compared to the wild-type. Growth and heart defects were not affected by the sex of the embryo.

7. All figures, including supplementary should have the y-axes starting at 0 – which is better for identifying the magnitude of the change. They should all include information on the number of litters used and employ litter averages for statistics (eg Supp Figure 3D is unclear).

Figures were edited accordingly. The information about the number of litters was added to the figure legends. As requested, we adjusted y-axis to start at 0 in all graphs (Figure 4F, Figure 4—figure supplement 2A and Figure 5-figue supplement 1A). In the figure legend for Figure 4—figure supplement 1D, we added requested information: ‘Embryonic and neonatal body weight of the wild-type (blue) and △H19/+ (green) samples at E11.5, E12.5, E14.5, E17.5, E18.5 and PN0 (mean ± SD). 6 litters for E11.5, 6 litters for E12.5, 3 litters for E14.5, 11 litters for E17.5, 3 litters for E18.5,14 litters for PN0 are presented. (E) Body weight of +/△H19 neonates (mean ± SD). 3 litters are presented.’

8. Text stating Data not shown: Please show all data or do not describe it.

As requested, we have removed/replaced four ‘data not shown’ in the previously submitted documents (at indicated Line) as follows:

Line 121: We deleted ‘In contrast to the liver where the paternal hIC1 did not lead to significant

abnormal morphology (data not shown),’.

Line 177: We deleted ‘and data not shown’. (see line 196)

Line 195: We replaced ‘Figure 4D and data not shown’ with ‘see Figure 4D for example’. (see lines 214-215)

Line 378: We replaced ‘data not shown’ with ‘Figure 6—figure supplement 5. (see line 398 and Figure 6—figure supplement 5)

9. Using in silico/bioinformatic analysis, are the authors able to estimate what proportion of the DEGs may be directly (vs indirectly) modified by the H19 ncRNA in the relevant mutants?

Because our mouse models are not linear with respect to *H19* overexpression, it is difficult to make a direct correlation between the level of *H19* expression and which genes are altered. However, data in Figure 6—figure supplement 2 and 3, where we compared *+/hIC1, △H19/hIC1,* and *△H19/+* endothelial cells to identify the genes whose expression is solely affected by the increased *H19*, suggest that *H19* affected a certain group of DEGs such as *Fgf10* or the imprinted gene network, independently of the loss of *Igf2* (also see lines 281-307).

10. I realise it is out of the scope of the paper, but it would have been amazing to have undertaken similar transcriptomic analyses of the placenta at E12.5 – to assess how the placenta and heart may similarly or differentially respond to the H19/Igf2 gene alterations. Moreover, undertaking placental or cardiac-specific gene manipulations would inform further on the nature of the plac-heart axis and how it may be mediated through changes in the H19/Igf2 imprinted locus. Descriptions of these future studies should be included in the discussion.

We agree with the reviewer and appreciate the insight. The suggestion was reflected on our discussion and a future experiment would be to profile transcriptomic changes in placental cells of our mouse models, as reflected in the following added text: “Transcriptomic analysis of fetoplacental endothelial cells from our mouse models would help us understand the role of *H19/Igf2* in placental development. Moreover, generating a tissue-specific *hIC1* mouse model, if possible, would allow us to determine the causal relationship between cardiac and placental phenotypes.” (see lines 420-423).

References

Esquiliano, D. R., Guo, W., Liang, L., Dikkes, P., & Lopez, M. F. (2009). Placental Glycogen Stores are Increased in Mice with H19 Null Mutations but not in those with Insulin or IGF Type 1 Receptor Mutations. *Placenta*, *30*(8). https://doi.org/10.1016/j.placenta.2009.05.004

Hur, S. K., Freschi, A., Ideraabdullah, F., Thorvaldsen, J. L., Luense, L. J., Weller, A. H., Berger, S. L., Cerrato, F., Riccio, A., & Bartolomei, M. S. (2016). Humanized H19/Igf2 locus reveals diverged imprinting mechanism between mouse and human and reflects Silver-Russell syndrome phenotypes. *Proceedings of the National Academy of Sciences of the United States of America*. https://doi.org/10.1073/pnas.1603066113

Lengner, C. J., Camargo, F. D., Hochedlinger, K., Welstead, G. G., Zaidi, S., Gokhale, S., Scholer, H. R., Tomilin, A., & Jaenisch, R. (2007). Oct4 Expression Is Not Required for Mouse Somatic Stem Cell Self-Renewal. *Cell Stem Cell*, *1*(4). https://doi.org/10.1016/j.stem.2007.07.020

Lopez, M. F., Dikkes, P., Zurakowski, D., & Villa-Komaroff, L. (1996). Insulin-like growth factor II affects the appearance and glycogen content of glycogen cells in the murine placenta. *Endocrinology*, *137*(5). https://doi.org/10.1210/endo.137.5.8612553

Sakata, Y., Kamei, C. N., Nakagami, H., Bronson, R., Liao, J. K., & Chin, M. T. (2002). Ventricular septal defect and cardiomyopathy in mice lacking the transcription factor CHF1/HEY2. *Proceedings of the National Academy of Sciences of the United States of America*, *99*(25). https://doi.org/10.1073/pnas.252648999

Tanaka, M., Hadjantonakis, A. K., Vintersten, K., & Nagy, A. (2009). Aggregation chimeras: Combining es cells, diploid, and tetraploid embryos. *Methods in Molecular Biology*, *530*. https://doi.org/10.1007/978-1-59745-471-1_15